# Illness Narrative Master Plots Following Musculoskeletal Trauma and How They Change over Time, a Secondary Analysis of Data

**DOI:** 10.3390/bs14111112

**Published:** 2024-11-19

**Authors:** Andrew Soundy, Maria Moffatt, Nga Man (Nicole) Yip, Nicola Heneghan, Alison Rushton, Deborah Falla, Lucy Silvester, Nicola Middlebrook

**Affiliations:** 1School of Sport, Exercise and Rehabilitation Sciences, University of Birmingham, Birmingham B15 2TT, UK; lenicoy@outlook.com (N.M.Y.); n.heneghan@bham.ac.uk (N.H.); 2School of Allied Health Professionals and Nursing, University of Liverpool, Liverpool L69 3BX, UK; m.moffatt@liverpool.ac.uk; 3School of Physical Therapy, Western University, London, ON N6A 3K7, Canada; arushto3@uwo.ca; 4Centre of Precision Rehabilitation for Spinal Pain, School of Sport, Exercise and Rehabilitation Sciences, University of Birmingham, Birmingham B15 2TT, UK; d.falla@bham.ac.uk; 5Institute for Applied & Translational Technologies in Surgery, University Hospital Coventry, Coventry CV2 2DX, UK; lucy.silvester@uhcw.nhs.uk; 6Department of Health Professions, Faculty of Health and Education, Manchester Metropolitan University, Manchester M15 6BX, UK; n.middlebrook@mmu.ac.uk

**Keywords:** illness narratives, master plots, quest, chaos, restitution, trauma, qualitative

## Abstract

Introduction; to the best of the authors knowledge, no past research has established how illness narrative master plots are expressed initially and then if and how they change longitudinally following musculoskeletal trauma. The aim of the present research was to consider how specific master plots were expressed, interact, and change across time following musculoskeletal trauma. Methods: A narrative analysis was undertaken that included individuals who had experienced a musculoskeletal traumatic injury. Individuals were included if they were an inpatient within 4 weeks of the first interview, had mental capacity to participate, and were able to communicate in English. Three interviews were undertaken (within 4 weeks of injury, then at 6- and 12-months post-injury). A 5-stage categorical form—type narrative analysis was performed. Results: Twelve individuals (49.9 ± 17.5 years; 7 male, 5 female) completed interviews at three time points following the trauma event (<4 weeks, 6 months, and 12 months). Three main narrative master plots appeared to work together to facilitate a positive accommodation of the trauma event into the individual’s life. These included the resumption narrative, the activity narrative, and the quest narrative. Finally, less often regressive narratives were identified, although these narratives were, at times, actively avoided. Discussion: The current results provide important consideration for how narratives are used within clinical practice, in particular the value of how these three narratives could be accessed and promoted.

## 1. Introduction

Musculoskeletal trauma refers to life—changing or life-threatening injuries caused by mechanisms of energy [1]. In addition to this, it is accompanied by deranged physiology and requires significant medical treatment that normally includes admission to a hospital intensive care unit, the provision of blood products, and surgical intervention [2]. There are around 22,000 cases of musculoskeletal trauma in the United Kingdom each year [3]. Past research has focused extensively on the physical aspects of the injury, whilst more recently, research has focused on the significant psychosocial aspects that accompany the event. For instance, past research has identified that up to 50% of patients can experience depression and post-traumatic stress disorder, and over a third of patients can experience anxiety [4]. The level of psychopathology appears to significantly and negatively influence physical recovery initially following the injury [5] as well as years after the injury [6]. Pain also has been observed for years following the trauma and has been associated with worse health outcomes, like anxiety and depression [7].

Following musculoskeletal trauma, patients require rehabilitation that can account for their physical, psychological, and social needs [8]. During rehabilitation, attention should be given to the psychological and social concerns of patients to improve their physical health [9,10]. Health care professionals should know how to provide psychological support [11] and emotional support [12]. Knowing how to navigate expectations and fears of the future is particularly important and challenging [13]. One way of considering this is by paying attention to individuals’ lived experience [14].

Illness narratives provide an accessible way to understand lived experience. Illness narrative master plots represent patient experience by focusing on a general storyline of that experience; the storyline is recognised by an underlying plot. Illness narrative master plots can provide a statement about living with illness that is time-orientated and reveals important aspects of psychology, including acceptance and hope [15]. Understanding more about the master plots could pave the way for psychosocial interventions to be developed, something that is important following musculoskeletal trauma, [10]. Three main master plots [16,17] following the onset of life—changing illness have emerged as dominant in the literature. These are the chaos narrative master plot (which illustrates a sense of hopelessness for the future), the quest narrative master plot (which seeks to embrace the future), and the restitution narrative master plot (which focuses on being restored to one’s past). How and if these narrative master plots are expressed following musculoskeletal trauma is largely unknown, although research has been able to demonstrate that expressions do exist. For instance, review-based research by Norris et al. [14] identified that following musculoskeletal trauma, individuals identify the importance of getting back to ‘normal’, which may represent a restitution narrative. However, there is a need to understand this further, as past research has identified that the restitution narrative is less frequently observed when a severe type of disability has resulted [16]. Thus, knowing how the restitution narrative is expressed or if a variant is expressed is important to establish. In a similar way, acceptance of the unchangeable nature of trauma has been identified as important following trauma [14] and this could provide a basis for the expression of the quest narrative master plot illustrating embracement or more regressive narratives illustrating loss. For instance, a variant of the quest narrative master plot (quest auto-mythology) is possible [16,17] and may be associated with expressions that represent a change in the character [14]. In contrast, the experience of trauma can be associated with a grieving process and the loss of roles and relationships [18]. This may be reflected in sad or tragic narrative master plots [15].

It is important that further research identify what narrative master plots exist, as this information can be used to inform clinical practice and develop a better clinical environment that is positive for the patient [15]. It is also important that research consider how this expression changes following the onset of trauma, as there are psychological and emotional needs of patients that vary across time and that are impacted by specific events during recovery [12,18]. To the best of the authors’ knowledge, no past research has considered how the expression following musculoskeletal trauma is represented by illness narrative master plots. Thus, the aim of this study is to provide an understanding of the expression of narrative master plots following a traumatic event and identify how these plots change across time.

## 2. Materials and Methods

This study is reported according to the Standards for Reporting Qualitative Research [19]. A study protocol was developed and can be accessed for more specific details of procedures undertaken [20].

### 2.1. Qualitative Approach and Research Paradigm

The research approach undertaken was a secondary analysis of data, which used a narrative approach. This narrative approach was situated within the social constructivist paradigm with a relativist ontology and subjectivist epistemology. Within this approach, each person’s view is equally valid. A secondary analysis of data is highly valuable where data for specific groups is hard to obtain [21]. However, secondary analysis of data must be associated with the original data collected to make it meaningful, and this is considered a challenge [21]. The original interview schedule provided access to concepts that are considered central to understanding narrative master plots. These central concepts included participants’ expressions of recovery from the past towards the future, the perceived ability to psychologically adjust to the traumatic experience, and the ability to express hope for the future.

### 2.2. Researcher Characteristics

Two researchers (NM, MM) undertook all interviews.

### 2.3. Context and Setting

Interviews were conducted face-to-face and online with individuals.

### 2.4. Sampling and Sample Size

A convenience sample of individuals from two major trauma centres in the UK was invited to take part. The sample size was identified as requiring enough participants to establish different narrative master plots. The current study sought to establish transferability [22] of the master plots, which are well established across groups, rather than the unique stories and experiences from which they are observed. In past research, small numbers. such as 8 individuals, have been used to represent specific master plots [23]. One of the reasons for this lower number is likely due to the self-selected nature of the sample and, to some extent, views that are aligned and more common. For this research, we were interested in all the participants who completed interviews at three time points. This meant that a sample of 12 was possible. This was identified as sufficient to establish details of the most common narrative master plots and identify some form of transferability of findings to other settings.

### 2.5. Ethical Approval

Prior to interviews starting, individuals were informed that all interviews were voluntary, they could stop at any point, and they could choose not to answer any questions they didn’t wish to. Please see below for the ethical review statement.

### 2.6. Data Collection Methods and Instruments

Interviews were undertaken at three time points. Time point 1 was within 4 weeks of injury. Time point 2 represented 6 months following injury, and time point 3 represented 1 year after injury. The rationale for this was to consider and compare expressions from early to later stages of recovery and rehabilitation.

A topic guide was developed for the interviews by a research team based on past literature [24,25] and based on the classification of function, disability, and health (ICF) domains [26].

### 2.7. Data Analysis

Data analysis was undertaken using a 5-stage categorical form-type narrative analysis [27]. The lead author undertook the secondary analysis of all interviews. Stage 1 included identification of master plots from scanning the interviews, together with factors that may be influencing the master plots or critical moments from the interview that appeared to shape the master plots. Stage 2 provided a summary of master plots for all 3 time points for everyone. Stages 3 and 4 involved focusing on identifying specific narrative master plots and clear identification of any new master plots. Stage 5 involved identification of how the master plots were to be presented.

### 2.8. Techniques to Enhance Trustworthiness

Andrew’s [28] framework was used to enhance the quality of this paper. Quality was established by ensuring evidence for existing as well as new master plots. Further to this, interview scripts were examined for evidence of multi-layered stories and variants of the stories, as well as looking for negative cases or aspects of the data that appear untold.

## 3. Results

### 3.1. Demographics

A total of 12 participants (49.9 ± 17.5 years; 7 male, 5 female) completed all three time points. Eight participants had experienced multiple fractures, and four had experienced single fractures. The average Injury Severity Score (ISS) was 9.5 ± 3.7. For full demographics, please see Table 1.

### 3.2. The Narrative Master Plots

Three specific narrative master plots were identified, including the resume narrative, the action narrative, and the quest narrative. Finally, there were also some, but limited expressions, that represented regressive narratives. Table 2 provides example quotes for each narrative and time point.

#### 3.2.1. The Resume Narrative Master Plot

The basic plot of this narrative master plot was—I will return or resume activities, relationships, roles, or social identities that I previously assumed. The goal of this narrative was to look forward and (re)establish aspects of life that were identified as meaningful and ‘normal’ before the onset of the traumatic event. The narrative had a firm hope that was more open to change or problems than being considered concrete and definite. The idea of normal often would include being recovered or as good as new, whilst some would identify a focus around being fixed like new; a greater focus was on being able to access past meaningful activities. Given this, the restoration of the body and function was important, but complete physical restoration represented only one way to access what was considered normal and meaningful. Many times, resumption would focus on the (re)establishment of activities, like walking, roles, for instance within the workplace or relationships, where (re)establishing independence was important. The way this narrative master plot was expressed was unique to the individual and changed around what was represented as ‘normal’ for each participant.

##### Time Point 1

The resumption narrative at this time point most often reflected a desire to return to normal. ‘Normal’ represented activities, roles, or movement. For instance, getting out with children (P10), getting back to work (P1), having an adapted role at work (P8), walking and not being in a wheelchair (P7). Some individuals would attach short time frames to achieving these activities and roles, often expressed as months. For instance, ‘4 to 6 months to fully recover’ (P20) or ‘a couple of weeks’ for walking (P1). Resumption of activities was not always assumed; this was due to the recognition of the extent of the injury being life—changing (P8), that the extent to which activities could be achieved was unknown (P12), or that healing was still needed (P14). Some were aware that recovery could go wrong (P17) or that specific aspects of recovery could go wrong, for instance, that the bone grafting may still go wrong (P15). Further to this, others expressed being more open to the idea that change and resumption could occur more slowly (P8; P10). Some individuals focused on being restored (reflecting a restitution master plot); this was expressed as being as ‘good as new’ (P4) or being recovered (P20).

##### Time Point 2

The resumption narrative had progressed at this time point as some had resumed normal activities, roles, and relationships. For instance, driving a car or taking the stairs (P10). Some expressed resumption as a percentage that illustrated the number of normal activities, roles, or relationships that were considered resumed. This could be as high as being 90% recovered (P17) or 80–90% recovered (P20, P24). An extended time frame for resuming activities was given compared to time point 1. For instance, 6 months’ time for walking (P7) or being in a good place (P20), a year to hit targets (P10), or 1 or 2 years to get back to normal living (P1, P4). One individual didn’t give a time scale but believed the injury would be fixed (P17), reflecting a restitution narrative. Although others (P14) identified being fixed as not possible (anti-restitution), but rather focused on being able to resume walking. Activities and roles not established yet were identified, such as not being able to run (P18), or this was expressed as goals to be achieved, such as going to the gym (P1) or running 5k and swimming (P24). For some, the resumption narrative was less certain for several reasons, including uncertainty around healing (P10, P24) or because of the advice given from health care professionals (P15). Resumption stories of others were referenced by some to contrast uncertainty and identify possibility for eventually being able to access normal (P1, P7, P10). Examples included people who were back to ‘normal’ (P8) or 20 to 30 years after the accident (P1, P10, respectively). In one instance, this contrasted with what health care professionals had told them (P1).

##### Time Point 3

The resumption narrative had progressed at this time point as many had resumed normal activities, roles, and relationships; however, the narrative was presented alongside more specific limitations. Limitations included: flexibility and muscle strength (P17), going up a ladder or doing netball (P18), running after the kids (P24), experiencing pain when performing certain activities like jumping from a height or climbing the stairs (P4). For some, the resumption of more difficult activities was possible, such as working in a workshop (P7), hoovering the car, and cutting toenails (P17). Others had progressed less but would still reference ‘getting normality back’ (P15) or having normality creeping in (P10). Still others would identify an extended timeframe for when they expected resumption of activities to begin. P14 identified that a scan demonstrated why the resumption of activities was not possible, and P7 identified that the importance of resumption of activities was relative, as ‘being old he didn’t worry so much about image’.

#### 3.2.2. The Activity Narrative Master Plot

The basic plot was that the action or activity I take in the present will provide me with progress towards a better future. In contrast, the narrative explicitly did not associate it with looking backwards or dwelling on the past and loss. The narrative represented an opportunity to gain independence, and that had the potential to aid an aspect of health and well-being. Action represented an act of engaging in a process, treatment, or activity following which made a future more possible. The activity identified within the narrative was vast but could include medication, surgery, accessing support, adhering to rehabilitation, physical activity, exercise or sport, or functional movements that could provide a basis for hope and change. Most often, it was represented by referring to movements and rehabilitation. It provided a source of hope when an action or activity was identified in the future, like surgery, and was affirmed or enhanced by moments of progress. Limits to action were identified when activities attempted failed or were identified as limited because of a concern around the risk of performing the activity.

##### Time Point 1

At this time point, the activity narrative master plot was focused on the need to act to make small gains or take small steps to achieve goals around independence. For example, this included getting better and being determined to push through (P12), seeing everything as a challenge and trying to accomplish it rather than dwelling on what had happened (P1). Specific movements, however, could be identified as not possible, like taking a shower or getting dressed (P1, P12).

Moments were identified as helping the activity narrative. These moments included looking forward to the day of surgery (P8) and achieving small changes as something that was positive and motivating (P10, P17). Specific moments of change and progress were identified as motivating and allowed for more goals to be identified; This included standing up (P17), walking (P1), being able to move up the stairs, and getting a wheelchair (P18). In contrast, moments of failure or limitations of activity were identified, this included attempting the stairs (P1) and not being allowed to do more than a specific activity like toe touch weight bearing (P18). One participant (P17) identified being more risk adverse following the traumatic event and identified that this limited the activity that was attempted.

##### Time Point 2

During this time point, the action narrative had evolved to reflect activities and actions that allowed the individual to solve specific problems and maintain independence (P2; P14). For instance, this included getting up the stairs using hands and a ‘bum shuffle’ (P14). The narrative was supported by a strong and continued need for independence (P14). The ability to access the narrative was supported by named characteristics, for instance, having a tenacity and determination to keep going (P10). The activity narrative could be enhanced by specific moments that allowed action. For instance, this could include the prosthetic fitting (P7) or waking up and not feeling in pain (P17). Moments of adverse outcomes could result from activity; this included experiencing pain (P17, P24) or having tried action and fallen over (P7). Some negative outcomes like pain could be seen as positive, e.g., pain because muscles have worked hard (P4).

Limits of action were linked with the risk of undertaking movements or activities that could create or cause fear and potentially impact progression. Danger was identified from performing tasks like standing on a stall or dropping from a height (P10), walking and ‘catching myself out’ (P1), undertaking activities, and damaging the joint (P15; P18). Awareness of pain and danger from movements like bending or writing of heavy lifting (P17), using pain as an indicator to ‘take a step back’ (P1) or as indicating the body is being pushed too far (P14), fatigue, which could add to concerns felt around action and movement (P8). Alternatively, P24 identifies an awareness of danger when driving, which is how the accident occurred previously.

##### Time Point 3

At this time point, many were able to identify that action and activity worked and needed to be continued. This would allow them to be stronger than before the accident (P4, P18), walk longer than before (P14), and make life easier in the future (P7). It would also allow a return to sport (P18) or exercise classes (P24). Action allowed some to focus on specific moments in the future; this more often occurred where improvements were perceived to be associated with the action taken (P10, P12, P17). For instance, this could be crutches-free (P10). If improvement was perceived even slightly, it provided a path for further possibilities (P7, P15, P24). Pain was a primary limiter of action, as it stopped actions (P17). For instance, expressions around pain included not being able to take a step without pain (P8); others identified that pain prevented goals being set (P20). Some accepted pain as part of action (P8), but others identified a concern that pain could be associated with the named activity or action forever (P24). Knowing the experience of pain prevented action, this could be attempting a particular movement (P17) or undertaking an activity like driving for a long time (P24). The experience of pain could be complicated by knowing that pain often didn’t give a warning, and the level of pain could not be controlled (P8).

#### 3.2.3. The Quest Narrative

The basic plot for the quest narrative was that if I can accept the problem or challenge that I face, take a break from it, get around it, make the most of it, or look at the positive aspects left, then I can embrace the present circumstances and continue to engage in life and overcome that which is faced. Thus, whatever challenge was faced, individuals telling this narrative were able to see the situation as an opening or opportunity and could consistently identify a valued future as well as identifying gains from the experience.

##### Time Point 1

At this time point, a general quest narrative was most often identified. The plot of the narrative illustrated the importance of reframing. For instance, this could include not seeing themselves as disabled but rather ‘extremely able’ (P14). For others, the plot was illustrated by a need to work with what is possible. For instance, if their knee hurt, they would work with the upper body (P1). Some identified the need to get around problems. For instance, getting around on crutches (P12). As part of such solutions, some expressed an acceptance that accidents can happen (P10) and, if they do, consequences are known (P14). Two individuals (P18, P20) identified a change of character or a quest auto-mythology. This included prioritising health, including changing nutritional habits and eating better, sticking to professional advice, and undertaking rehabilitation activities and exercise.

##### Time Point 2

The quest narrative at this time point focused on accommodating changes. This included continuing activities by planning what to do (P15), being aware of the environment and how to navigate it, this could include an awareness of curbs on the pavement when using a wheelchair outside (P7). Some identified the importance of accepting that what they regarded as normal (pre-accident engagement in activities, functions, and relationships) may not return (P8). Others focused on considering what was possible and would make the best of what was possible (P24) or work with those activities that were possible (P14).

##### Time Point 3

The quest narrative most often focused on valuing what was possible as a major aspect of the plot. This included focusing on what was left, maximizing the present (P14), and being proud of being disabled (P7). This may reflect an auto-mythology narrative and a change in character. Others identified benefit finding as an outcome following the accident. For instance, this could include meeting people who would not have met otherwise (P14). Other narratives focused on finding a different way forward (P1) or adapting to challenges. For instance, this could include identifying the best shoes to wear (P1) or knowing when to use crutches (P15). For P7, health care professionals helped the quest narrative, saying there is nothing more that can be done (P7).

Problems with the health system were recounted by three individuals (P7, P14, P17). This identified problems with the systems used for support and getting access to or receiving care, and could have led to a quest manifesto, although there was no demand for social change from individuals as they expressed empathy towards the pressured environment experienced by health care professionals within the National Health Service.

#### 3.2.4. Regressive Narratives and Moments

The basic plot of a regressive narrative was that life was not progressing as planned, the future was unlikely to change because recently progress had appeared to be limited, plateaued, or got worse, and the likelihood of change was not possible. This could be impacted by medical assessments and experiences over time. The expression illustrates elements of a sad narrative plot most often, with a more regressive tragic plot less often. The expression often identified that the possibility of change was out of the patient’s control, and distance was created between where the individual was at present and where they hoped that change could lead in the future. Some participants reflected on time—limited regressive or sad moments between interviews, stating that during these experiences they were low, sad, or depressed.

##### Time Point 1

Regressive narratives did not appear within this time point. This was likely due to time since injury and the dominance of the other narrative expressions and ability to be hopeful.

##### Time Point 2

At this time point, regressive narratives and plots could be associated with the tragic narrative where individuals identified that nothing more can be done (P1) or that there are no more goals to achieve (P24). Some identified uncertainty for the future, which questioned a more progressive narrative. For instance, this could include not knowing if the foot will heal or get better (P15). One individual (P12) identified a more regressive narrative due to an additional injury that had prevented progress and meant more time was needed before progression could be achieved.

##### Time Point 3

The regressive master plot at this time point was often focused on limited improvement. Some had experienced a plateau after some earlier improvement (P4); others had taken a step backwards at a particular time point (P24). P12 talked about having an additional problem of a catheter and that before that was removed, he was ‘down; and ‘not doing very well’. Others identified that changes were no longer possible (P10, P24) for them.

Some regressive moments were presented as things to avoid. For instance, not dwelling on loss (P1) and that being negative can make living harder (P18). However, moments of assessment like scans could illustrate bad news (P17). A more permanent sad narrative was expressed by P8, identifying the process of recovery as grieving, and this expression fitted more with a tragic narrative and may be more permanent.

## 4. Discussion

To the best of the authors’ knowledge this is the first study to consider the interplay of narrative master plots across time for individuals following a major musculoskeletal trauma. The main narrative master plots identified in the current study were able to work together to allow a continued positive outlook. The resumption narrative and the desire to resume meaningful activities, roles, and relationships were most often associated with the action narrative and individuals being able to identify specific activities and actions taken to achieve aspects of normality. The quest narrative master plot enabled individuals to have continued access to the resumption narrative master plot when limits or restrictions were identified. These narratives provide an illustration of how the current participants in the study were able to switch between master plots to help maintain a positive, future—focused outlook. This is something that is considered important for health outcomes in people following trauma [29] and people who face chronic and palliative conditions [15]. There appeared to be limited evidence of consistent or dominating regressive narratives. Regressive narrative master plots appeared most often within specific historical moments and situations.

### 4.1. The Resume Narrative Master Plot

The patient’s perceived need to set and achieve meaningful goals throughout recovery was supported by past literature related to trauma [14] and life—changing events [30]. The resumption narrative supported this through the desire to return to what are considered by the individuals in the study, normal and meaningful activities, relationships, and roles. The resumption narrative has a critical advantage over the restitution narrative in that it will likely be associated with less judgement or concern from a therapist. For instance, past literature has identified that therapists can identify the restitution narrative as unrealistic and illustrating the patients cannot accept what is happening [31]. Further to this, therapists can also be concerned about phrases like wanting to be ‘as good as new’ [16]. Across time, the resumption of normality around activities and roles becomes a focus for patients in the current study. One reason for this appeared to be that more specific limitations became known and there appeared to be no expression of the restitution by time point 3. It is, therefore, important for therapists to understand that the restitution may well be naturally challenged across time, and to prevent it prematurely could cause more distress than benefit.

Time point 2 appeared to be the most associated with expressions of uncertainty towards the future. Also, during this time point, the time frame for recovery was extended compared to time point 1. Onaway individuals appeared to establish a more hopeful and open outlook was by identifying positive stories of others who, often years following trauma, had regained normality. This is supported by other literature that identifies that having access to positive stories can provide pathways for hope to develop [15,32]. Further to this, review evidence has identified that narrative interventions are able to impact an individual’s psychological well-being by increasing the motivation for action, providing meaning and fulfilment, enhancing a perception of quality of life, and influencing perceptions of control [33]. These factors are important as they will likely influence the individual’s ability to return to work and undertake normal activities. Both factors are regarded as essential goals to achieve following trauma [34].

### 4.2. The Action Narrative

The action narrative master plot appeared to play a pivotal role in helping individuals find hope and a way forward. This finding is supported by existing theoretical understanding [15] but also by literature that has identified the high number of patients (77%) supplementing rehabilitation and the perception that this will influence therapeutic outcomes [35]. Patients in the current study valued making small, progressive steps towards improvement. These steps of progression are regarded as important landmarks indicating progress towards a sense of normality [14]. Identifying different action—related pathways to achieving goals is important for individuals who experience trauma. For instance, past literature has identified this as being able to use adaptive equipment, pacing, taking rest, and changing forms of activity [18]. Undertaking action or activities was likely a process by which hope was generated, and a focus on action likely protected against other more regressive narratives. This is supported by past literature, which has consistently identified a negative association between hope and depression [36].

Action appeared to be a more independent activity at later time points, and individuals were able to specify limits of action and had to learn to deal with adverse outcomes following action like pain and reduced energy. It may be that at later time points individuals require greater agency thinking (thoughts about oneself that produce confidence and energy to being and continuing to act towards a goal) as particularly important [37]. Individual determination and self-motivation are critical in generating this type of thinking [18]. At time point 2, specific limits of action were identified, risks of action were considered, and activities that could not be performed became clear to participants. Past research identifies the need to manage fear avoidance, notably at 6 or 12 months where concerns are specifically related to undertaking movement [11]. Interestingly, a fear of reinjury has previously appeared as a limiting factor 3 years following the trauma [18]. Further, a common fear for people following trauma is how long is needed for healing to take place, since this perception, if negative, influences their ability to access activities [34]. Thus, individuals may benefit from an assessment around fear or movement following discharge and strategies to help reduce fear of movement.

Pain was expressed more specifically at time point 3 with specific movements and identified as something that may not change in the future. An understanding of this by therapists could be useful, and patients may benefit from psychological treatments associated with acceptance.

### 4.3. The Quest Narrative

The quest narrative appeared to be used as an expression that aided patients in overcoming obstacles, challenges, and pain. The narrative appeared to provide a way forward where action was not possible to create change, and this helped support the resumption narrative. It is likely that being able to embrace challenges via the quest narrative enhanced an individual’s goal—related energy and enabled the resumption narrative to continue [38]. The quest narrative master plot could be aided by past experiences of accidents and challenges and moments of reality when individuals were told about limits of progress evident or rehabilitation potential.

### 4.4. Regressive Narrative and/or Moments

Often moments of sadness, depression, or time reflecting on the tragic nature of the events were actively avoided. By using the above narratives during the first two time points, individuals stated they would not dwell on the loss and change. At time point two, one participant’s narrative was dominated by a regressive narrative. This was influenced by a lack of perceived improvement and less progression with their recovery than hoped for, and the belief that the current experiences of pain and the limitations of movement would likely continue. In past literature, these experiences have been shown to limit the ability to be hopeful [15] and are also linked with a perception that control has been lost [34]. Importantly, individuals who have expressed a narrative of despair or chaos most often felt like their life was out of control and were also dependent on others [17]. Additionally, individuals expressing this type of narrative identify an empty present and a future that is lost to the effects of the illness [39]. Limited improvements after discharge and a less positive outlook can create a lack of positivity in patients following discharge [29]. Supporting patients during regressive moments is considered an important element of psychological and emotional wellbeing support following musculoskeletal trauma [12] or following significant loss and challenge [15].

### 4.5. Limitations

The self-selected sample used for this study may have reduced narrative expressions within the results, and further research is required to consider this. As such, a greater understanding of regressive narrative master plots may be required in future research. The lack of demographic data collected for individuals in the current study may limit the context for individual master plots. Further work that considers particular groups and cultures is needed to elaborate on the current findings.

## 5. Conclusions

This research has identified an important association between narrative master plots across time. The self-selected nature of the sample may be one reason for the consistency in the narrative presented and what allowed individuals to keep looking forward positively. Further research is needed to confirm these findings.

## Figures and Tables

**Table 1 behavsci-14-01112-t001:** Providing the demographics of participants.

Participant ID	Gender	Age	Fracture	Single Fracture or Multiple Fracture	Soft Tissue Injury	Further Detail	ISS Score	ISS Mild/Mod/Severe
1	Male	34	Yes	Multiple	Y	1. Closed Right Femoral Fracture 2. Closed left tibial fracture 3. Left foot injury 4. Right supraorbital laceration 5. Hand Injury	10	Moderate
4	Female	35	Yes	Single	N	1. Left diaphyseal femoral shaft fracture	9	Moderate
7	Male	57	Yes	Single	N	1. Proximal tibial fracture 2. Through knee amputation	16	Major
8	Male	59	Yes	Single	Y	1. Right open elbow dislocation 2. Laceration of brachial artery 3. Distal radius fracture	9	Moderate
10	Male	44	Yes	Multiple	N	1. Bilateral shoulder dislocation 2. Right shoulder fracture with TSR 3. Left femoral fracture	9	Moderate
12	Male	80	Yes	Multiple	Y	1. Left hip dislocation 2. Left acetabular dislocation 3. Left distal humerus fracture 4. Left olecranon fracture 5. Left 5th rib fracture 6. Left knee laceration 7. Left grade 3 PCL/MCL injury	5	Mild
14	Female	73	Yes	Multiple	N	1. Left tibial plateau fracture 2. Fractured ribs	8	Mild
15	Male	36	Yes	Multiple	Y	1. Right and left medial malleolus fracture 2. Left open calcaneus fracture 3. Left tibial lacertation 4. Elbow injury	9	Moderate
17	Male	60	Yes	Multiple	N	1. Left rib fracture 2. Left clavicle fracture 3. T11 and L1 fracture	9	Moderate
18	Female	20	Yes	Single	N	1. Right femoral shaft fracture 2. Knee ligament injury	9	Moderate
20	Male	57	Yes	Multiple	N	1. Right scapula fracture 2. Left comminuted tibia 3. Fibula fracture	4	Mild
24	Female	44	Yes	Multiple	N	1. Right acetabulum 2. Right hip dislocation 3. Left tibial plateau 4. Right rib fracture 9–12 posterior 5. Right transverse process lumbar spine	17	Major

**Table 2 behavsci-14-01112-t002:** Provides illustrations of quotes from participants.

Narrative Master Plot	Time Point	Example Quotes
Resumption	1	‘*a couple of weeks [time]… if I can walk before Christmas I’ll be laughing*”. (P1) “*I just know it’s going to take a while, but, like, I’ll be back good as new at some point*” (P4) “I want to get back on a prosthetic and I want to be walking, I want to be everybody else’s height or whatever height I was” (P7)
	2	“*[I am at present] about 80% function may be now, and I’m hoping to get back to 90%”* (P24) “*people that have been told they are never going to walk again. Next thing they’re winning an Olympic gold medal*”. (P1) “[I have watched] *above knee people… and I think… if they can do that… ride a bike, ride a motorbike, climb mountains [I can]*”. (P7)
	3	“*I was going to be okay by Christmas. It’s now been a year. I’m obviously still not okay. So, that’s probably the biggest wakeup. So, I’ve times that by about 10 [years*]”. (P4) “*could see [from the scan] that the other leg was just as bad as that one. So, it looks like a minefield of bones”* (P14)
Action	1	“*I have been back in to see the consultant this week, on Tuesday, but again, I had to do that myself and get myself in the car. We had to drag me into the car and drive me to [hospital location]*”. (P24) “*[I] did my stairs, and then got a wheelchair back. So that was… Thursday was my turning point; my good day*” (P24) stating “*[I] did my stairs, and then got a wheelchair back. So that was… Thursday was my turning point; my good day*”. (P18)
	2	states “*woke up one morning and the ribs were fine… so I thought well, I’m not in pain now*”. (P17) *I’ve had a few incidents of crying in the car and things, especially if the kids have been fighting and I haven’t felt just safe*” (P24)
	3	“*I’ll practice that more and more and more. I know it takes time, but it does seem to work*” (P14) *“can be a bit psychologically difficult to deal with because you’re sort of asking yourself, is this something I’m going to have to keep forever?”* (P24)
Quest	1	“*other people might think of me as disabled. I think I’m extremely able. Yeah. It’s a matter of attitude, isn’t it?*”. (P14) “*I am aware that that is maybe not possible, but I’m trying to sort of keep quite a neutral mindset, not worry one way or the other because I don’t think that that’s helpful*”. (P10) *“I just feel lucky to be alive you know and its completely changed my life, my attitude, you know I just want to look after myself more. I’ve lost, my weight is down, I’ve never been so… my weight has never been so low*”. (P20)
	2	“*in my mind I was going this way, my accident made me go this way [a different way], but I still want to keep on going forward but just on a different way”* (P1) “*now I have to plan everything and plan my food, my weekly intake of food… its been hard adjusting to all that sort of thing*”. (P15)
	3	“*She [health care professional] said to me jokingly, she goes, look, you’ve got to realise that that’s not your own leg, it’s a prosthetic leg… I needed someone to tell me that because sort of like, the way things were going, it was like, everybody was saying, yeah, it will be good as new*”. (P7) *my arm… I’ve lost all muscle in it… but it will be alright… I’m a bit restricted with that but at least I can get about*” (P12)
Regressive	1	
	2	“*[I] try to keep some positive thoughts about it [recovery]. At the same time… I do get thoughts of like I’m not going to be to walk again. Is my foot ever going to get better or will it need an amputation in the years ahead”.* (P15)
	3	“*I feel as though if I sat down and dwelled in all it, it’s just going to make it worse*”. (P1) “*I really started to do really well with my initial recovery, then I just kind of like plateaued, almost went backwards. That was really hard*”. (P4) “*it’s sort of comeback worse I suppose. I thought I was over the hump and more accepting and that’s difficult now because I think the realisation, the reality is that we’re pretty much as good as it will get. And in some ways, it’s like worse. My back is really painful*”. (P8)

## Data Availability

The Appendix A contains anonymous data.

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
