# Peer review of "Illness Narrative Master Plots Following Musculoskeletal Trauma and How They Change over Time, a Secondary Analysis of Data"

_behavsci, 2024, doi:10.3390/bs14111112_

Round 1

Reviewer 1 Report

Comments and Suggestions for Authors

The manuscript employed a 5-stage categorical narrative analysis to explore the longitudinal expression, interaction, and changes of illness narrative master plots following musculoskeletal trauma. The focus of the analysis was on the resumption, activity, and quest narrative master plots, along with regressive narratives, observed at three different time points: within 4 weeks of injury, 6 months, and 12 months. The results demonstrate the expression of each narrative master plot over time. Notably, the resumption, activity, and quest narrative master plots appeared to work synergistically, conveying a positive attitude. Additionally, regressive narratives were identified, highlighting the perception that change was beyond the control of the patients.

Here are the comments provided:

  1. It is important to note that a convenience sample may introduce selection bias and may not accurately represent the general population. For instance, patients with a more positive attitude might be more likely to participate in the interviews.
  2. Clarification is required regarding the "Single/Multiple" column in Table 1. If it represents single fracture and multiple fractures, there are 8 patients experiencing multiple fractures and 4 patients with single fractures, instead of 7 and 5.
  3. If possible, it would be beneficial to include additional demographic information, such as education level and marital status, as these variables may influence the expression of illness narrative master plots.
  4. Since a 5-stage categorical form type narrative analysis was utilized, it would be valuable to present results specific to each stage. For example, what are the identified factors influencing the master plots or critical moments in stage 1?
Comments on the Quality of English Language

The English language used in this manuscript is clear and comprehensible. However, in section 3, it would be beneficial to provide a more concise summary of the results, incorporating relevant statistics. Instead of referring to "some individuals" and "others," it would be more effective to specify the number of patients out of the total 12 who exhibited a particular expression. Additionally, to enhance clarity, it is recommended to summarize the findings regarding the expression, interaction, and changes of illness narrative master plots at the outset of section 3.2.

Author Response

Thank you for your comments and points made. Below we have responded to the concerns and comments.

  1. It is important to note that a convenience sample may introduce selection bias and may not accurately represent the general population. For instance, patients with a more positive attitude might be more likely to participate in the interviews.

AS: We have added this as a limitation at the end of the discussion.

  1. Clarification is required regarding the "Single/Multiple" column in Table 1. If it represents single fracture and multiple fractures, there are 8 patients experiencing multiple fractures and 4 patients with single fractures, instead of 7 and 5.

AS: Thank you for this comment and observation. Changes have been made to reflect this typographical error.

  1. If possible, it would be beneficial to include additional demographic information, such as education level and marital status, as these variables may influence the expression of illness narrative master plots.

AS: This data was not requested. This has been added as a limitation.  

  1. Since a 5-stage categorical form type narrative analysis was utilized, it would be valuable to present results specific to each stage. For example, what are the identified factors influencing the master plots or critical moments in stage 1?

AS: Thank you for this comment. An audit trail for the analysis and each stage has been created. This is provided in a supplementary file.

Reviewer 2 Report

Comments and Suggestions for Authors

This is an important, patient-focused paper. A lot of work has gone into recording, coding and interpreting the patients’ post-trauma narratives, over three time points. Some interesting questions arise, such as how available narratives are for modification, that will certainly be addressed in future research.

For a paper about text analysis, however, the grammar is surprisingly poor – particularly with respect to punctuation, noun-verb agreement, word compounds, apostrophes and incorrectly included or omitted words. Simply put, scientific manuscripts in the archival literature should be written in grammatical English, and this paper is not.

Some of the needed edits are listed below, but not all are included.

Solutions are – the authors reading ‘Fowler’s Modern English Usage’ and applying the rules therein, or - asking an older native English speaker who studied grammar as part of their school curriculum to carefully edit the manuscript.

SPECIFIC EDITS

l.17 authors’ knowledge (needs possessive apostrophe)

l.19 aim of the present research was to (omit able)

l.23 interview, had

l.32 could be accessed

l.46 appears to significantly

l.60 the storyline is recognised

l.61 time-oriented

l.69 narrative master plots

l.71 review-based

l.77 way, acceptance

l.91 authors’ knowledge, no past

l.92 is represented (noun-verb agreement)

l.110 participants’ expression

l.119 was invited (noun-verb)

l.121 different narrative

l.121 plots. The current

l.127 who completed

l.136 Time point 1

l.144 form-type

l.178 plot was – I will return

l.200 extent……..was unknown

l.219 were identified, such as

l.231 included; flexibility

l.241 move 3.2.2 to LH margin

l.243 contrast, the

l.252 surgery, and

l.260 possible, like

l.287 danger was identified from

l.298 be being crutches-free

l.301 others identified

l.310 that I face

l.318 this could include

l.328 changes. This

l.332 on considering

l.339 who would

l.346 manifesto, although

l.367 more can be done

l.385 authors’ knowledge

l.394 narratives provide an illustration of how the

l.404 what are

l.408 and therapists

l.429 forward. This

l.432 making small progressive

l.443 individuals

l.445 and continuing

l.451 Interestingly,

l.453 if negative,

l.464 an individual’s

l.465 could be aided

l.475 In past literature, these experiences

l.477 with loss of perception

l.480 Additionally, these individuals

Comments on the Quality of English Language

For a paper about text analysis, however, the grammar is surprisingly poor – particularly with respect to punctuation, noun-verb agreement, word compounds, apostrophes and incorrectly included or omitted words. Simply put, scientific manuscripts in the archival literature should be written in grammatical English, and this paper is not.

Author Response

Reviewer 2

AS: Thank you for your comments and constructive feedback.

This is an important, patient-focused paper. A lot of work has gone into recording, coding and interpreting the patients’ post-trauma narratives, over three time points. Some interesting questions arise, such as how available narratives are for modification, that will certainly be addressed in future research.

For a paper about text analysis, however, the grammar is surprisingly poor – particularly with respect to punctuation, noun-verb agreement, word compounds, apostrophes and incorrectly included or omitted words. Simply put, scientific manuscripts in the archival literature should be written in grammatical English, and this paper is not.

Some of the needed edits are listed below, but not all are included.

Solutions are – the authors reading ‘Fowler’s Modern English Usage’ and applying the rules therein, or - asking an older native English speaker who studied grammar as part of their school curriculum to carefully edit the manuscript.

AS: Thank you for this observation. The main problem here is dyslexia of the corresponding author. I have re-gone over the manuscript by listening to it and I have asked other author to consider and check. I appreciate yo

SPECIFIC EDITS

l.17 authors’ knowledge (needs possessive apostrophe)

l.19 aim of the present research was to (omit able)

l.23 interview, had

l.32 could be accessed

l.46 appears to significantly

l.60 the storyline is recognised

l.61 time-oriented

l.69 narrative master plots

l.71 review-based

l.77 way, acceptance

l.91 authors’ knowledge, no past

l.92 is represented (noun-verb agreement)

l.110 participants’ expression

l.119 was invited (noun-verb)

l.121 different narrative

l.121 plots. The current

l.127 who completed

l.136 Time point 1

l.144 form-type

l.178 plot was – I will return

l.200 extent……..was unknown

l.219 were identified, such as

l.231 included; flexibility

l.241 move 3.2.2 to LH margin

l.243 contrast, the

l.252 surgery, and

l.260 possible, like

l.287 danger was identified from

l.298 be being crutches-free

l.301 others identified

l.310 that I face

l.318 this could include

l.328 changes. This

l.332 on considering

l.339 who would

l.346 manifesto, although

l.367 more can be done

l.385 authors’ knowledge

l.394 narratives provide an illustration of how the

l.404 what are

l.408 and therapists

l.429 forward. This

l.432 making small progressive

l.443 individuals

l.445 and continuing

l.451 Interestingly,

l.453 if negative,

l.464 an individual’s

l.465 could be aided

l.475 In past literature, these experiences

l.477 with loss of perception

l.480 Additionally, these individuals

AS: Thank you for identifying the above typographical errors. We have provided a clean and marked version so you can consider the changes made.